# Thermal and Interfacial Stability of PPS-Fabricated Segmented Skutterudite Legs for Thermoelectric Applications

**DOI:** 10.3390/ma18132923

**Published:** 2025-06-20

**Authors:** Mirosław J. Kruszewski

**Affiliations:** Faculty of Materials Science and Engineering, Warsaw University of Technology, 141 Wołoska Str., 02-507 Warsaw, Poland; miroslaw.kruszewski@pw.edu.pl

**Keywords:** skutterudite, thermoelectric, joints, aging, stability, interface evolution, pulse plasma sintering

## Abstract

The development of thermoelectric modules based on skutterudite materials requires stable, low-resistance interfaces between segments operating at different temperature ranges. This study investigates the microstructure, thermoelectric performance, and thermal stability of the following two joints: In_0.4_Co_4_Sb_12_/Co_4_Sb_10.8_Te_0.6_Se_0.6_ (*n*-type) and CeFe_3_Co_0.5_Ni_0.5_Sb_12_/In_0.25_Co_3_FeSb_12_ (*p*-type), fabricated by pulse plasma sintering (PPS). Scanning electron microscopy (SEM) and energy-dispersive spectroscopy (EDS) analyses confirmed the formation of well-bonded interfaces without pores or cracks. Aging at 773 K for 168 h did not result in morphological or chemical degradation, and phase composition remained unchanged according to X-ray diffraction (XRD). Surface Seebeck coefficient mapping and contact resistance measurements showed negligible changes after annealing, confirming electrical stability. To provide context for potential applications, theoretical energy conversion efficiencies were estimated based on measured thermoelectric properties, yielding 13.2% and 10.1% for the *n*- and *p*-type segmented legs, respectively. Additionally, measured coefficients of thermal expansion (CTE) indicated low mismatch between jointed materials, supporting good mechanical compatibility. The results demonstrate that the selected material combinations are thermally, chemically, and electrically stable and can be effectively used in segmented thermoelectric legs for intermediate-temperature applications.

## 1. Introduction

Thermoelectric (TE) materials have been increasingly studied due to their ability to directly convert waste heat into electricity via the Seebeck effect. This offers a clean and efficient means of energy recovery, especially valuable in transportation, manufacturing, and space applications where thermal gradients are common [1]. By exploiting temperature gradients, thermoelectric generators (TEGs) can provide clean, maintenance-free, and scalable energy conversion systems.

Among TE materials, skutterudites stand out for their excellent combination of high electrical conductivity and intrinsically low thermal conductivity, especially when modified via element filling or substitution [2]. These features allow skutterudites to reach competitive thermoelectric performance, with a dimensionless figure of merit (ZT) exceeding 2.0 in some *n*-type compositions [3,4] and over 1.4 in optimized *p*-type systems [5].

The performance of these materials is quantified by the dimensionless figure of merit, ZT, which is defined as $\mathrm{ZT}=\frac{\alpha^{2}\sigma }{\lambda }T$, where α is the Seebeck coefficient, σ is the electrical conductivity, T is the absolute temperature, and λ is the thermal conductivity [6]. In practice, a high ZT over a wide temperature range is desired for efficient energy conversion. However, many materials only perform well in narrow temperature windows, limiting their overall efficiency [7].

To overcome this limitation, segmented thermoelectric legs have been developed, which combine two or more materials optimized for different temperature regions. This segmentation ensures that each material operates within its most efficient range, improving average ZT and energy conversion efficiency [8]. Several studies have validated this strategy experimentally, reporting significantly improved module performance through careful material pairing [9,10].

One recent study by Wan et al. constructed fully segmented skutterudite modules and reached efficiencies exceeding 10%, emphasizing the practical potential of such architectures [11]. Similarly, efforts by Prado-Gonjal et al. and Ochi et al. confirmed that segmented modules can be reproducibly fabricated, while maintaining mechanical integrity under operational loads [9,10].

More recent investigations further support the advantages of segmentation. Bao et al. demonstrated the fabrication of stable *n*-type segmented joints between skutterudite and half-Heusler materials using low-temperature bonding, showing excellent interfacial properties and thermal stability after aging [12]. In addition, Chen and Chiou applied evolutionary computation to optimize segmented leg geometries based on skutterudites, achieving a predicted energy conversion efficiency of up to 14.05%—highlighting the importance of structural design alongside material selection [13].

These findings underscore not only the thermoelectric potential of segmented architectures but also the critical role of interfacial stability, geometric optimization, and long-term reliability in high-performance module design.

Nonetheless, high performance alone is insufficient. The interfaces between different thermoelectric segments must maintain thermal, mechanical, and chemical stability throughout extended thermal cycling [14,15]. If thermal expansion mismatches or interfacial degradation occur, the device may fail prematurely [16,17]. As Zhu et al. emphasized, synergy between materials is critical—not just for efficiency but also for long-term reliability [15].

Thermo-mechanical compatibility is especially important in skutterudite systems, which can suffer from phase instability, elemental segregation, or oxide formation at high temperatures [18,19]. Strategies to mitigate these problems include minimizing thermal expansion mismatch and optimizing diffusion bonding techniques [20].

Effective consolidation methods, such as pulse plasma sintering (PPS), spark plasma sintering (SPS), or other current-assisted methods, have gained attention as reliable techniques for fabrication and joining thermoelectric materials. These approaches enable rapid densification and bonding with reduced grain growth [21,22].

While SPS has been extensively studied in the context of skutterudite consolidation and joint fabrication [18,19], the application of PPS remains largely unexplored in the literature. PPS, developed and applied at the Warsaw University of Technology, uses short high-current pulses to initiate rapid sintering, offering unique advantages in controlling microstructure and preserving interface integrity during joining. A detailed comparison of PPS and SPS may be found in [23].

Although some results exist for bulk TE materials obtained via PPS [23,24,25,26,27,28], no comprehensive studies to date have addressed segmented thermoelectric legs joined by PPS, particularly under long-term thermal aging conditions. Critical issues, such as elemental diffusion, interfacial degradation, and thermomechanical compatibility at segment boundaries, remain poorly understood in this context.

This study addresses this gap by investigating segmented *n*-type (In_0.4_Co_4_Sb_12_/Co_4_Sb_10.8_Te_0.6_Se_0.6_) and *p*-type (CeFe_3_Co_0.5_Ni_0.5_Sb_12_/In_0.25_Co_3_FeSb_12_) thermoelectric legs fabricated using PPS. We analyze their microstructure, thermoelectric transport, chemical stability, and mechanical compatibility before and after prolonged annealing at 773 K. To our knowledge, this is the first detailed experimental demonstration of durable, segmented skutterudite legs joined via PPS.

## 2. Materials and Methods

### 2.1. Sample Preparation

For the fabrication of segmented thermoelectric joints, two *n*-type and two *p*-type skutterudite-based compositions were selected.

*n*-type: Co_4_Sb_10.8_Te_0.6_Se_0.6_ and In_0.4_Co_4_Sb_12_*p*-type: In_0.25_Co_3_FeSb_12_ and CeFe_3_Co_0.5_Ni_0.5_Sb_12_

High-purity starting materials were used, as follows: cobalt (powder, 99.98%, ~2 μm), antimony (ingot, 99.999%), selenium (shots, 99.999%, 2–6 mm), tellurium (pieces, 99.999%), nickel (powder, 99.9%, −325 mesh), indium (ingot, 99.999%), cerium (ingot, 99.9%), and iron (powder, 99.9%, 1–9 μm). Each mixture included a 2 at.% excess of Sb to compensate for volatilization.

All synthesis operations were carried out in a glovebox under high-purity argon atmosphere (O_2_ and H_2_O ≤ 0.5 ppm) to minimize contamination. The powder mixtures were sealed under vacuum (≤5 × 10^−3^ Pa) in carbon-coated quartz tubes for heat treatment.

Annealing procedures were as follows:For In_0.4_Co_4_Sb_12_, In_0.25_Co_3_FeSb_12_ and CeFe_3_Co_0.5_Ni_0.5_Sb_12_: the samples were heated to 1323 K at 2 K/min, held for 3 h, and water quenched.For Co_4_Sb_10.8_Te_0.6_Se_0.6_: the sample was heated in two steps—first to 923 K at 1 K/min for 3 h, then to 1073 K at 1 K/min for another 3 h, followed by water quenching.

All samples then underwent a second annealing stage at 873 K for 168 h (heating rate 3 K/min), after which they were cooled inside the furnace. The obtained ingots were ground into powder for further processing.

### 2.2. Fabrication by Pulse Plasma Sintering

The powder materials were consolidated and joined using the PPS technique. This process was carried out in a vacuum atmosphere (≤5 × 10⁻^3^ Pa) under a uniaxial pressure of 50 MPa.

The PPS parameters were as follows:*n*-type segments and joints: sintered at 923 K for 10 min*p*-type segments and joints: sintered at 873 K for 10 min

The PPS technique enabled the solid-state bonding of the thermoelectric segments without the use of solders or interlayers. In addition to the fabrication of segmented legs, bulk samples of each material were consolidated under the same PPS conditions for the purpose of measuring thermoelectric properties, thermal conductivity, and thermal expansion coefficient (CTE).

The PPS enables the rapid, solid-state consolidation of thermoelectric segments without the use of interlayers or soldering agents. Due to its short sintering times and localized heating, this technique minimizes microstructural degradation and interfacial diffusion. Its application in this study allowed for the direct fabrication of both segmented legs and reference bulk samples under identical thermal and mechanical conditions.

After sintering, the bulk and joined samples were cut into smaller sections appropriate for microstructural observations and physical property measurements.

### 2.3. Aging Procedure and Characterization

Thermal aging tests were conducted in sealed quartz ampoules under vacuum at 773 K for 168 h to assess thermal stability. This duration (168 h) was selected based on commonly adopted practice in thermoelectric reliability studies, where short-to-medium-term aging protocols are used to evaluate the early-stage microstructural and chemical stability of joints under thermal stress. Several recent studies have demonstrated that this time scale is sufficient to reveal interfacial degradation, elemental diffusion, and crack formation, while remaining below the threshold for bulk material degradation [12,29,30].

The microstructure and elemental distribution at the joints were investigated using a Hitachi SU-70 scanning electron microscope (SEM) equipped with an energy-dispersive spectroscopy (EDS) system (Tokyo, Japan). Phase composition was examined using X-ray diffraction (XRD) with a Bruker D8 Advance diffractometer (Billerica, MA, USA) and Cu Kα radiation. The Seebeck coefficient α and electrical conductivity σ were measured using the standard four-probe method in vacuum. Thermal diffusivity was evaluated using laser flash analysis (LFA, Netzsch, 457 MicroFlash, Selb, Germany). The density ρ of the consolidated samples was measured using the Archimedes’ principle. Thermal conductivity λ was calculated according to the formula λ = α∙C_p_∙ρ, where C_p_ is the theoretical heat capacity. The linear CTE was determined using a dilatometer (DIL 402 SE, Netzsch, Selb, Germany) under an argon atmosphere. Local distributions of the Seebeck coefficient were measured at room temperature using a scanning thermoelectric microprobe (STM) across a 2 × 0.5 mm area at the joint cross-section, with a spatial resolution of 0.05 mm. Electrical resistivity distribution was recorded using a custom-built apparatus with a resolution of 0.01 mm, also at room temperature.

## 3. Results and Discussion

### 3.1. Microstructure and Chemical Characterization

The quality of the joint interfaces plays a crucial role in the performance and reliability of thermoelectric modules. In this section, we present the microstructural evaluation of the fabricated joints based on SEM observations and elemental analysis using EDS. The aim was to assess the structural integrity of the bonded regions, identify possible interfacial reactions or secondary phases, and determine the elemental distribution across the joints.

As shown in Figure 1 and Figure 2, the interfaces of the PPS-fabricated In_0.4_Co_4_Sb_12_/Co_4_Sb_10.8_Te_0.6_Se_0.6_ and CeFe_3_Co_0.5_Ni_0.5_Sb_12_/In_0.25_Co_3_FeSb_12_ joints are bonded without any cracks or pores, indicating that the consolidation process parameters were well optimized.

For the In_0.4_Co_4_Sb_12_/Co_4_Sb_10.8_Te_0.6_Se_0.6_ joint, the interface is not readily distinguishable in SEM images. Even elemental mapping using EDS does not clearly resolve the boundary between the two materials. However, in the Se elemental map, some indication of the interface is observed, where impurity phases appear restricted to the lower part of the image. Due to overlapping elemental peaks, precise interface identification remains challenging.

For the CeFe_3_Co_0.5_Ni_0.5_Sb_12_/In_0.25_Co_3_FeSb_12_ joint (Figure 2), SEM imaging revealed contrast variations at the interface region, which were attributed to the presence of minor cerium oxide impurities. A detailed chemical composition analysis further confirmed a distinct boundary between the two bonded thermoelectric materials. The performed EDS line scans provided additional insight into the elemental distribution at the interface, reinforcing the stability and compatibility of the selected materials for thermoelectric applications.

To further investigate the interfacial region, an EDS line scan analysis was conducted (Figure 3). This method proved to be more effective in delineating the boundary between the two materials based on the variation in dopant concentrations. The line scan analysis conducted for In_0.4_Co_4_Sb_12_/Co_4_Sb_10.8_Te_0.6_Se_0.6_ joint (Figure 3a) confirmed the presence of impurity phases containing Se, Te, and In. The detection of minor secondary phases enriched in In, Se, and Te at the interface suggests localized dopant segregation phenomena, commonly observed in skutterudite systems doped with chalcogens and indium. Khovaylo et al. [20] reported the formation of InSb-rich regions along grain boundaries in In-filled CoSb_3_, resulting from overfilling during synthesis. Similarly, Visnow et al. [21] demonstrated that even minimal deviations in indium content can lead to the precipitation of InSb, with strong implications for microstructural stability. In our study, despite the presence of secondary phases, no degradation of the joint integrity was observed, suggesting that the segregation remains confined and non-detrimental to the mechanical properties of the segmented legs.

Figure 3b presents the results of the elemental line scan analysis conducted on the CeFe_3_Co_0.5_Ni_0.5_Sb_12_/In_0.25_Co_3_FeSb_12_ joint. The compositional profile across the interface enables a clear distinction between the two materials, marking the boundary of the joint.

The EDS line scan across the CeFe_3_Co_0.5_Ni_0.5_Sb_12_/In_0.25_Co_3_FeSb_12_ joint revealed the presence of a Ce-rich oxide phase in the interfacial region. This feature is most likely related to contamination or partial oxidation of the starting Ce material, which is highly reactive even under controlled synthesis conditions. Similar phenomena have been previously reported by Ballikaya et al. [31], who detected trace amounts of Yb_2_O_3_ in double-filled skutterudites and attributed them to minor oxygen uptake during processing. The presence of such oxides, while undesired, is not uncommon in rare-earth-based skutterudites and can indicate limited chemical homogeneity at the microstructural level.

The successful incorporation of Ce into the skutterudite structure in our sample is nonetheless evident from the remaining homogeneous matrix, which aligns with expectations for partially filled *p*-type skutterudites. According to Tan et al. [14], the formation of CeSb_2_ secondary phases occurs when the Ce content exceeds approximately 0.94 in Ce_y_Fe_3_CoSb_12_ compounds. In our case, the effective Ce content within the skutterudite structure appears to be below this threshold, which is consistent with the partial localization of Ce in oxide form. This suggests that the presence of the Ce oxide phase may reduce the actual filling fraction in the matrix, thereby altering the transport properties and potentially underestimating the impact of Ce as a filler in electrical behavior.

### 3.2. Phase and Thermal Stability of the Joints

The long-term functionality of thermoelectric materials depends on their thermal and chemical stability under operating conditions. This subsection discusses the chemical and phase composition of the joint regions in the as-fabricated state, as well as the results of post-annealing studies conducted at 773 K for 168 h. The goal was to evaluate potential phase transformations or interfacial degradation resulting from thermal exposure.

To evaluate the thermal stability of the fabricated joints and the potential evolution of the interfaces at high temperatures, an aging test was conducted at 773 K for 168 h (Figure 4 and Figure 5). The interfacial regions of both joints showed no visible changes in morphology in the SEM images, indicating high thermal stability. Additionally, no significant changes in the chemical composition of the joints were observed with elemental mapping.

To further investigate the interfacial region of the In_0.4_Co_4_Sb_12_/Co_4_Sb_10.8_Te_0.6_Se_0.6_ joint, an elemental line scan analysis was performed (Figure 6a). The obtained data allowed for a determination of the boundary between the two materials and confirmed the presence of impurity phases at the interface. The detected secondary phases appear similar in composition to those observed in the as-fabricated state. However, their apparent quantity in the line scan analysis is slightly higher than before annealing. Since no significant morphological changes were observed in the microstructure, this discrepancy is most likely attributed to the specific location of the line scan rather than an actual increase in impurity concentration. This interpretation is further supported by elemental mapping results, which indicate a comparable distribution of impurity phases before and after heat treatment.

The line scan analysis of the CeFe_3_Co_0.5_Ni_0.5_Sb_12_/In_0.25_Co_3_FeSb_12_ joint after annealing at 723 K for 168 h under vacuum conditions is presented in Figure 6b. The results confirm that no significant alterations occurred in the interfacial composition compared to the as-fabricated state. A distinct boundary between the two materials remains clearly visible, reinforcing the conclusion that the bonding between the materials is stable under the applied thermal conditions. This stability suggests that the interface maintains its structural integrity and does not undergo undesirable diffusion-driven degradation, making the material combination suitable for thermoelectric applications.

The detection of regions with Se, Te, or In at the interface suggests local dopant segregation, which is a common phenomenon in skutterudite-based materials doped with chalcogens. Dong et al. [17] demonstrated that, during high-pressure synthesis, Se and Te exhibit limited solubility in CoSb_3_ and tend to form secondary phases, such as Co(Sb_0.46_Se_0.54_)_2_. Similarly, Deng et al. [19] observed the formation of nanoscale InSb secondary phases in In/Ba/Te-doped CoSb_3_ systems prepared by HPHT, which remained stable without significant growth after thermal aging. In our case, the secondary phases detected by EDS line scan showed no morphological evolution or propagation during the aging treatment, indicating good chemical stability of the joint interfaces.

Furthermore, the presence of minor secondary phases without interfacial degradation aligns with observations made by Jiang et al. [18], who reported that Te–Ge double-substituted CoSb_3_ skutterudites retained excellent structural stability under high-temperature conditions due to a combination of point defects, dislocations, and grain boundary structures. This microstructural configuration promotes phonon scattering and thus supports both mechanical robustness and enhanced thermoelectric performance. Similarly, Battabyal et al. [22] reported that chalcogen doping in CoSb_3_ leads to local chemical inhomogeneities but enhances phonon scattering without structural destabilization.

It is worth noting that while these secondary phases may locally modify the interface composition, their limited volume fraction and high dispersion are unlikely to adversely affect the electrical or thermal transport properties of the joints. On the contrary, similar mechanisms were exploited by Kruszewski et al. [32], where highly homogeneous Se/Te-doped CoSb_3_ exhibited ultralow lattice thermal conductivity due to intensified phonon scattering without detrimental effects on structural integrity.

Therefore, the observed Se- and Te-rich regions at the interface are considered benign in the context of the overall joint performance, and they may even contribute positively to the thermal stability of the segmented legs.

XRD analysis was conducted to examine the phase composition of the fabricated thermoelectric joints before and after annealing at 773 K for 168 h under vacuum conditions (Figure 7).

For both *n*- and *p*-type joints, the diffraction patterns revealed the presence of the skutterudite phase as the dominant component in the as-fabricated materials. Additionally, a small fraction of antimony was detected, which is attributed to the excess added during the synthesis process to compensate for potential volatilization losses.

The annealing process did not induce any significant changes in the phase composition of the materials. The XRD patterns of both *n*- and *p*-type joints remained unchanged, indicating that the fabricated interfaces exhibit high phase stability under the applied thermal conditions. These results confirm that the thermoelectric joints maintain their structural integrity within the investigated temperature range.

The XRD analysis performed after the aging treatment did not reveal the formation of any new diffraction peaks, nor did it indicate measurable changes in peak positions or widths. This suggests that both segmented legs and interfacial regions maintained their phase integrity and crystalline structure after prolonged exposure to elevated temperatures. Such stability is of paramount importance for thermoelectric devices intended for long-term operation under thermal gradients.

Similar high-temperature phase stability has been observed by Dong et al. [17], who reported that Se/Te-doped skutterudites synthesized under high pressure retained their crystalline skutterudite structure even after extensive heat treatment, with only trace secondary phases remaining stable. Moreover, Jiang et al. [18] demonstrated that the introduction of Te and Ge into the CoSb_3_ matrix combined with HPHT processing led to stable microstructures characterized by point defects, dislocations, and multiscale grain size distributions, which remained unchanged after high-temperature cycling. These results are also consistent with observations made by Rogl et al. [33], who reported that InSb secondary phases formed during the preparation of In/Ba/Yb-filled CoSb_3_ remained stable and did not propagate or transform even after prolonged thermal exposure.

In our case, the lack of secondary compound formation at the interface region after thermal aging implies that both the filler composition and bonding process parameters were appropriately chosen to suppress detrimental interdiffusion or phase transformations. This aligns with previous findings by Kruszewski et al. [32], where simultaneous doping with Se and Te resulted in materials with ultralow lattice thermal conductivity and outstanding thermal stability due to the formation of defect-tolerant crystalline networks.

Altogether, our results confirm that the developed bonding approach ensures the structural robustness of the joint region and preserves the phase stability of the segmented legs during long-term operation, making the design highly suitable for advanced thermoelectric modules.

### 3.3. Thermoelectric Performance and Interfacial Properties

To ensure efficient energy conversion, the selected materials must exhibit not only high thermoelectric performance but also compatible electrical and thermal transport properties at the joint interface. This section presents measurements of the Seebeck coefficient, compatibility factor (CF), and contact resistivity, along with thermal expansion data. Together, these properties provide a comprehensive view of the electronic and thermomechanical behavior of the joints.

The thermoelectric properties of the materials used in this study are presented in Figure 8. The selected *n*- and *p*-type materials exhibit the highest ZT across different temperature ranges, which justifies their combination for thermoelectric module fabrication.

To maximize the average ZT parameter, the optimal joint temperatures for the thermoelectric joints were identified. The results indicate that the optimal temperature for the *n*-type joint is approximately 673 K, while for the *p*-type joint, it is around 623 K.

Furthermore, the compatibility factor defined as [11]:(1)$$\mathrm{CF}=\frac{\sqrt{1+\mathrm{ZT}}-1}{\alpha T}$$
where α is the Seebeck coefficient and T is the temperature, was calculated to evaluate the suitability of the selected materials for efficient power generation within the module (Figure 9). For optimal thermoelectric performance, the CF values at the interface should not differ by more than a factor of two over the relevant temperature range. The obtained CF values for the *n*-type joint are −2.69 and −2.84, indicating excellent compatibility. For the *p*-type joint, the CF values of 3.86 and 5.69 remain within the acceptable range.

Based on these findings, the combination of the investigated materials is deemed appropriate for thermoelectric applications, ensuring stable and efficient energy conversion.

To further evaluate the quality of the interfaces, a Seebeck scanning probe was used. Figure 10 presents the *n*-type joint in its as-fabricated state and after annealing at 773 K for 168 h under vacuum conditions. A slightly reduced Seebeck coefficient was observed in the interfacial region, which may be attributed to minor variations in chemical composition at the joint, potentially due to slight changes in doping concentration. However, after the annealing process, the thermoelectric properties remained largely unchanged, indicating that the heat treatment did not significantly affect the material’s performance. This suggests that the *n*-type joint maintains its thermoelectric stability within the investigated temperature range.

Similarly, the influence of thermal treatment on the *p*-type joint was analyzed. The surface distribution of the Seebeck coefficient before and after annealing exhibited negligible differences, confirming that the joint retains its thermoelectric properties. The results indicate that the *p*-type joint is thermally stable under these conditions, further supporting the reliability of the selected material combination for thermoelectric applications.

The electrical contact resistance of both *n*-type and *p*-type joints was evaluated in the as-fabricated state and after heat treatment at 773 K for 168 h under vacuum (Figure 11). The measurements were conducted in conjunction with surface Seebeck coefficient mapping to ensure the comprehensive characterization of the thermoelectric interfaces.

For both types of joints, no abrupt change in electrical resistivity was observed across the interface, regardless of the thermal treatment. This indicates that the bonding process yielded interfaces with homogeneous electrical behavior. The calculated contact resistance values at the interface positions are summarized in Table 1.

In both cases, a slight decrease in contact resistance was observed after annealing. This reduction may be attributed to improved atomic diffusion and the further relaxation of the interfacial region, which can enhance electrical contact by reducing microvoids or interfacial discontinuities initially present after fabrication.

According to the criteria proposed by Ryu et al. [34], contact resistance values in the order of 10^−6^ Ω·cm^2^ are indicative of high-quality interfaces, which have a negligible effect on the overall efficiency of thermoelectric devices. The measured values for both *n*- and *p*-type joints fall within this excellent range, demonstrating that the interfaces formed in this study are electrically well-matched and efficient.

The absence of degradation in electrical contact properties after heat treatment, combined with the favorable Seebeck coefficient distribution and material compatibility, suggests that devices fabricated using these materials can be expected to exhibit high efficiency and long-term stability in practical applications.

In thermoelectric modules operating under significant temperature gradients, mechanical reliability is strongly influenced by the mismatch in the CTE between adjacent materials. Differential thermal expansion may lead to interfacial stresses, delamination, or microcrack formation, particularly during repeated thermal cycling. Therefore, beyond electrical and thermoelectric optimization, ensuring thermal expansion compatibility is essential for long-term durability and functional stability.

The measured linear CTE was obtained for all investigated materials in the range of 310–723 K. The results are summarized in Table 2, which show relatively small differences between the components forming each joint. For the *n*-type leg, In_0.4_Co_4_Sb_12_ and Co_4_Sb_10.8_Te_0.6_Se_0.6_ exhibit CTEs of 10.6 × 10^−6^ K^−1^ and 11.7 × 10^−6^ K^−1^, respectively. In the *p*-type segment, the values are 12.0 × 10^−6^ K^−1^ for CeFe_3_Co_0.5_Ni_0.5_Sb_12_ and 10.5 × 10^−6^ K^−1^ for In_0.25_Co_3_FeSb_12_.

These differences, not exceeding 1.5 × 10^−6^ K^−1^ in either case, are considered moderate and within the acceptable range for direct bonding without buffer layers. The literature suggests that mismatches below ~2 × 10^−6^ K^−1^ are generally tolerated in thermoelectric joints without inducing critical residual stresses during thermal cycling [6,15,16]. This correlates well with the observed thermal stability of the interfaces in SEM and EDS analyses following the 168 h annealing tests. The selected material combinations used in this work inherently offer such compatibility without the need for additional diffusion barriers or interlayers, which simplifies the module architecture and may reduce the fabrication cost and complexity.

In summary, the measured CTE values, when combined with the results of the microstructural and electrical stability tests, strongly support the mechanical robustness and application potential of the studied thermoelectric joints in energy conversion devices operating across moderate-to-high temperature differentials.

### 3.4. Theoretical Conversion Efficiency

The theoretical energy conversion efficiency of thermoelectric modules is closely linked to the average ZT over the temperature range of operation. In this part, we present the calculated efficiency values for individual materials and segmented legs based on maximum and average ZT. The implications of segmentation on performance improvement, as well as the limitations of relying solely on peak ZT values, are also discussed with reference to relevant literature.

Table 3 presents the calculated values of theoretical energy conversion efficiency for the individual thermoelectric materials as well as for the assembled joints. The efficiency was estimated based on the following equation [35]:(2)$$\eta =\frac{T_{h}-T_{c}}{T_{h}}\cdot \frac{\sqrt{1+\bar{\mathrm{ZT}}}-1}{\sqrt{1+\bar{\mathrm{ZT}}}+\frac{T_{c}}{T_{h}}}$$
where T_h_ and T_c_ are the temperatures of the hot and cold ends of the thermoelectric leg, respectively, and ZT is the average dimensionless figure of merit over the operating temperature range. The average ZT values were obtained by averaging our measured ZT data over the 300–773 K range, consistent with the conditions applied during performance testing. This ensures that the estimated efficiency reflects the actual thermoelectric behavior of the fabricated materials. The values of T_h_ = 773 K and T_c_ = 300 K were used to reflect the typical operating conditions employed in our aging and transport measurements.

Among the investigated materials, In_0.4_Co_4_Sb_12_ and CeFe_3_Co_0.5_Ni_0.5_Sb_12_ demonstrated the most promising thermoelectric performance, with high values of both peak and average ZT. The theoretical conversion efficiency calculated for these materials individually was slightly below 13% for the *n*-type leg and approximately 10% for the *p*-type leg.

Importantly, by employing a segmented leg configuration—combining complementary thermoelectric materials within a single leg—it was possible to increase the average ZT and surpass the 13% and 10% efficiency thresholds for *n*- and *p*-type segments, respectively. This result highlights the practical advantage of segmentation in thermoelectric module design, particularly when the selected materials exhibit differing optimal operating temperature ranges. A similar strategy was adopted by Wan et al. [11], who demonstrated a full skutterudite-based segmented thermoelectric module reaching an efficiency of 10.4% by combining In_0.25_Co_4_Sb_12_ with Yb_0.35_Co_4_Sb_12_ (*n*-type) and Ce_0.9_Fe_3_CoSb_12_ with CeFe_3.85_Mn_0.15_Sb_12_ (*p*-type), fabricated via one-step sintering. Their study confirms that material pairing from a common chemical family not only improves thermal and mechanical compatibility but also leads to low-resistance interfaces and high device reliability.

It is also worth emphasizing the often-overlooked discrepancy between the ZT_max_ and the temperature-averaged ZT_avg_, which has direct implications for realistic efficiency predictions. For instance, in our case, Co_4_Sb_10.8_Te_0.6_Se_0.6_ exhibited a relatively high peak ZT, but its ZT_avg_ was significantly lower due to the narrow optimal temperature range. Relying solely on ZT_max_ for performance estimation can, therefore, lead to the substantial overestimation of the expected output in real operating conditions.

This distinction has been consistently discussed in the literature. Yusuf et al. [7] highlighted through numerical simulations that ZT_avg_ provides a more reliable indicator of actual energy conversion efficiency, especially for segmented or wide-range applications. They emphasized that optimized segmentation strategies should be based on compatibility of ZT_avg_ values, rather than ZT_max_ alone. This approach is further supported by experimental studies from Salvador et al. [8], where system-level efficiencies closely followed predictions based on ZT_avg_, not peak values. Our findings corroborate these conclusions, showing that segmentation—when designed with ZT_avg_ as the guiding parameter—offers a robust path toward high-performance thermoelectric devices.

Although a conservative temperature range of 300–773 K was used in this study to ensure material stability, recent reports suggest that extending the hot-side temperature by ~50 K can significantly enhance energy conversion efficiency. Several works have demonstrated that skutterudite-based modules operating at 823 K or higher exhibit improved thermoelectric performance and higher conversion efficiencies, typically in the range of 9–10% or above [29,30,36]. In particular, Wan et al. demonstrated a segmented skutterudite module reaching 10.4% efficiency at extended temperatures using a one-step sintering approach [11]. Furthermore, Chen and Chiou predicted up to 14.05% efficiency using optimized geometries for skutterudite-based segmented legs [13], while Bao et al. reported stable segmented joints with low interfacial resistance and thermal degradation under elevated temperatures [12]. While the calculated efficiencies of the segmented legs (13.2% for *n*-type and 10.1% for *p*-type) appear only slightly higher than for the individual materials, this is mainly due to the relatively narrow temperature range applied in this work. Segmentation is expected to deliver greater benefits when materials operate within their respective optimal temperature ranges, particularly at higher T_h_ values. Moreover, the benefits of segmentation extend beyond thermodynamic efficiency—it enables broader operating temperature ranges, improves mechanical compatibility, and reduces thermal mismatch at interfaces, all of which are critical for long-term stability in real device conditions. This indicates that the theoretical efficiencies reported in this work may be further improved with extended high-temperature operation, provided that chemical and mechanical stability is maintained.

It is important to note that the calculated ZT is a derived parameter, dependent on independently measured values of the Seebeck coefficient, electrical conductivity, and thermal conductivity. Each of these measurements carries its own experimental uncertainty, which propagates into the final ZT and efficiency estimates. Based on equipment specifications and the repeatability of the measurements, the estimated relative uncertainties are approximately ±3% for α, ±5% for σ, and ±7% for λ. This leads to an estimated combined uncertainty in ZT of approximately ±10–12%.

Accordingly, the reported theoretical energy conversion efficiencies of 13.2% (*n*-type) and 10.1% (*p*-type) for segmented legs should be interpreted with an associated uncertainty of approximately ±1–1.5 percentage points. Although this introduces some variability, the relative trends remain robust—segmentation consistently improves average ZT and conversion efficiency compared to individual materials within the bounds of experimental error.

## 4. Conclusions

The main findings of this study can be summarized as follows:Thermoelectric joints based on In_0.4_Co_4_Sb_12_/Co_4_Sb_10.8_Te_0.6_Se_0.6_ (*n*-type) and CeFe_3_Co_0.5_Ni_0.5_Sb_12_/In_0.25_Co_3_FeSb_12_ (*p*-type) were successfully fabricated using the pulse plasma sintering (PPS) technique for the first time.Microstructural characterization confirmed dense, crack-free interfaces without visible porosity. Minor impurity phases were observed by EDS mapping and line scans, but they did not compromise joint quality.Post-annealing tests conducted at 773 K for 168 h revealed the high thermal and chemical stability of the joints, with unchanged morphology and elemental profiles. XRD analysis confirmed phase stability.Seebeck coefficient mapping after thermal aging showed only minor variations, indicating preserved thermoelectric properties across the joint regions.Contact resistivity remained low before and after annealing, confirming the electrical stability of the interfaces. Compatibility factors calculated for the segmented joints were within acceptable limits.Thermoelectric performance evaluation showed high ZT values in both individual and segmented legs. Theoretical energy conversion efficiencies reached 13.2% (*n*-type) and 10.1% (*p*-type). Segmentation improved average ZT and thus enhanced overall efficiency.Measured coefficients of thermal expansion (CTE) showed minimal mismatch between the materials in each joint (<1.5 × 10^−6^ K^−1^), supporting good thermo-mechanical compatibility without the need for buffer layers.The selected material combinations and joining approach using PPS demonstrate strong potential for future integration into efficient, stable thermoelectric modules operating in the intermediate temperature range.

These findings demonstrate the viability of PPS as an effective, solder-free joining technique for fabricating segmented skutterudite thermoelectric legs with high thermal and electrical stability. To fully exploit the potential of the developed joints, future work should focus on integrating them into complete thermoelectric modules and evaluating long-term performance under real operating conditions, including repeated thermal cycling and mechanical stress. Additionally, exploring the scalability of PPS for larger devices and extending the hot-side operating temperature could further enhance system-level efficiency and durability.

## Figures and Tables

**Figure 1 materials-18-02923-f001:**
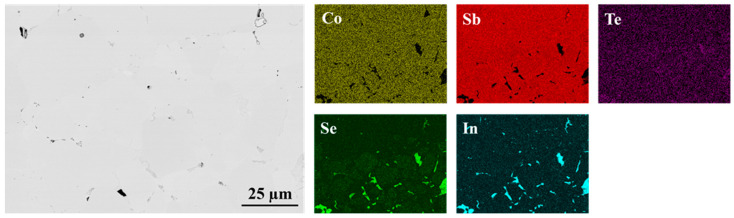
Interfacial microstructure of In_0.4_Co_4_Sb_12_/Co_4_Sb_10.8_Te_0.6_Se_0.6_ joint presented with elemental maps collected using EDS.

**Figure 2 materials-18-02923-f002:**
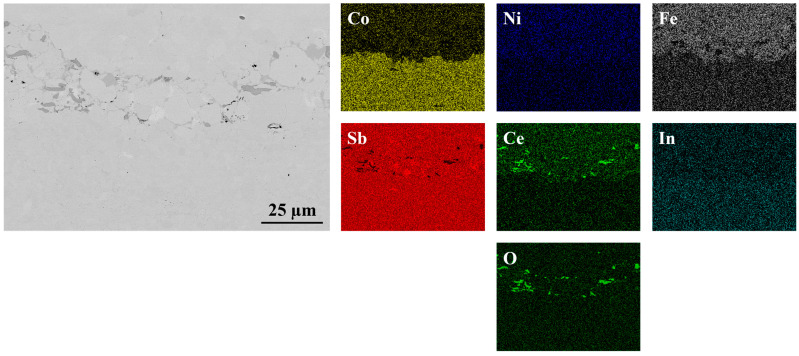
Interfacial microstructure of CeFe_3_Co_0.5_Ni_0.5_Sb_12_/In_0.25_Co_3_FeSb_12_ joint presented with elemental maps collected using EDS.

**Figure 3 materials-18-02923-f003:**
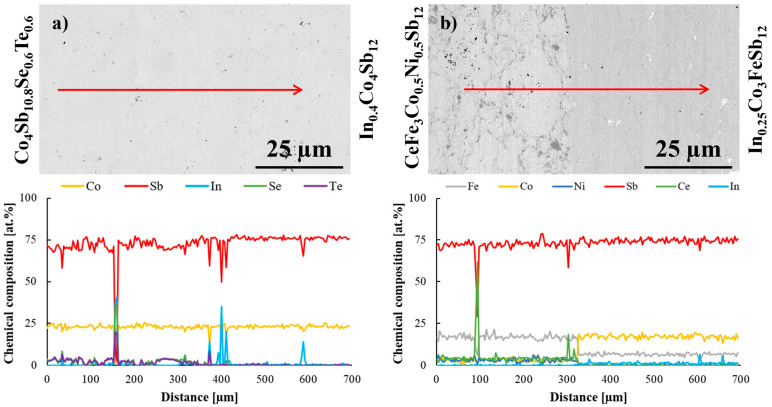
EDS line scan analysis of (**a**) In_0.4_Co_4_Sb_12_/Co_4_Sb_10.8_Te_0.6_Se_0.6_ and (**b**) CeFe_3_Co_0.5_Ni_0.5_Sb_12_/In_0.25_Co_3_FeSb_12_ joints. Arrows correspond to the analyzed lines.

**Figure 4 materials-18-02923-f004:**
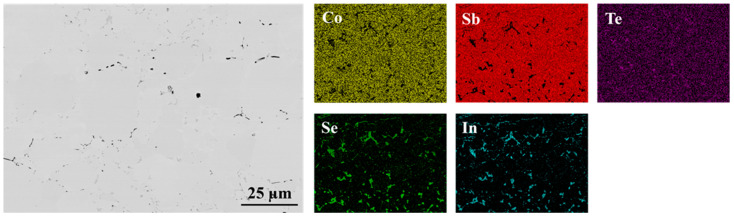
Interfacial microstructure of In_0.4_Co_4_Sb_12_/Co_4_Sb_10.8_Te_0.6_Se_0.6_ joint after the aging test at 773 K for 168 h presented with elemental maps collected using EDS.

**Figure 5 materials-18-02923-f005:**
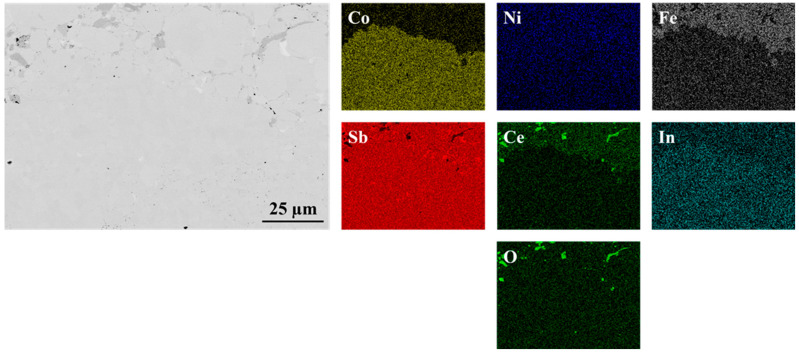
Interfacial microstructure of CeFe_3_Co_0.5_Ni_0.5_Sb_12_/In_0.25_Co_3_FeSb_12_ joint after the aging test at 773 K for 168 h presented with elemental maps collected using EDS.

**Figure 6 materials-18-02923-f006:**
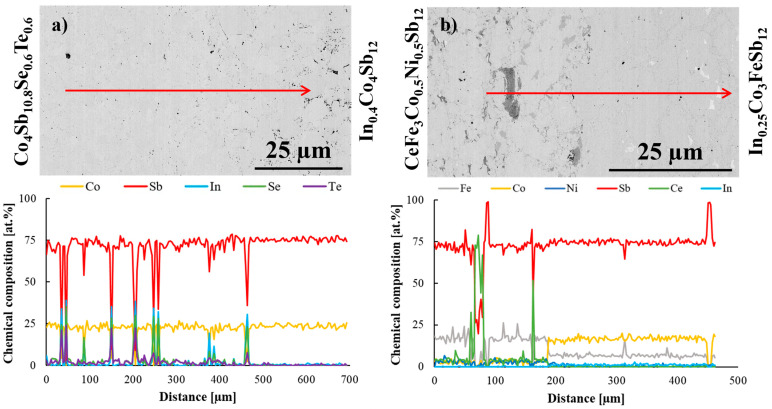
EDS line scan analysis of (**a**) In_0.4_Co_4_Sb_12_/Co_4_Sb_10.8_Te_0.6_Se_0.6_ and (**b**) CeFe_3_Co_0.5_Ni_0.5_Sb_12_/In_0.25_Co_3_FeSb_12_ joints after heat treatment at 773 K for 168 h. Arrows correspond to the analyzed lines.

**Figure 7 materials-18-02923-f007:**
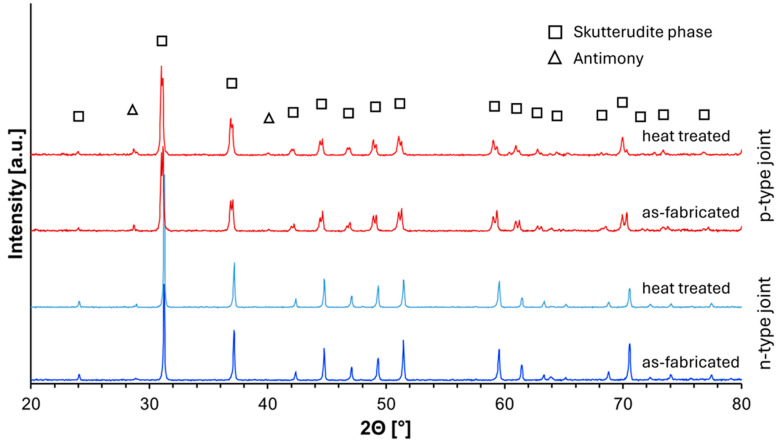
XRD analysis of In_0.4_Co_4_Sb_12_/Co_4_Sb_10.8_Te_0.6_Se_0.6_ (*n*-type) and CeFe_3_Co_0.5_Ni_0.5_Sb_12_/In_0.25_Co_3_FeSb_12_ (*p*-type) joint in as-fabricated and heat-treated state.

**Figure 8 materials-18-02923-f008:**
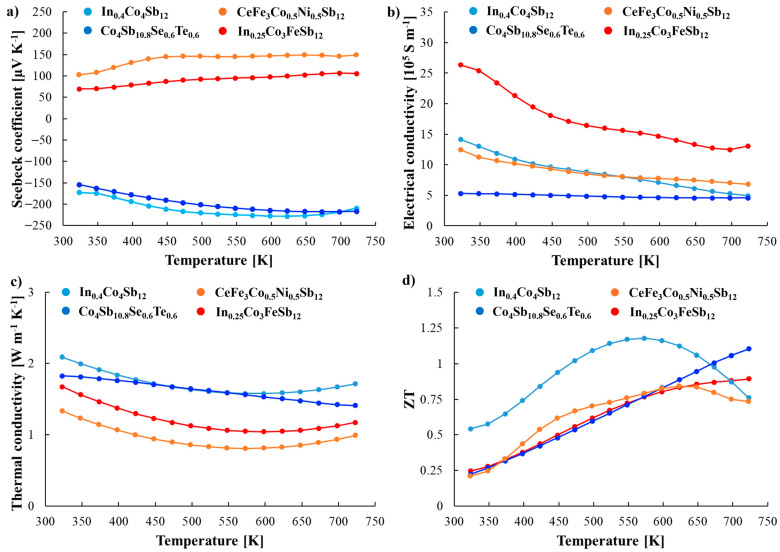
Temperature dependent (**a**) Seebeck coefficient α, (**b**) electrical conductivity σ, (**c**) thermal conductivity λ, and (**d**) figure of merit ZT of fabricated materials.

**Figure 9 materials-18-02923-f009:**
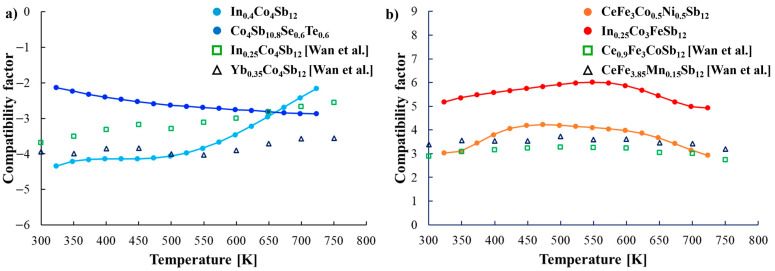
Temperature dependent (**a**) CF for *n*-type joint and (**b**) CF for *p*-type joint [11].

**Figure 10 materials-18-02923-f010:**
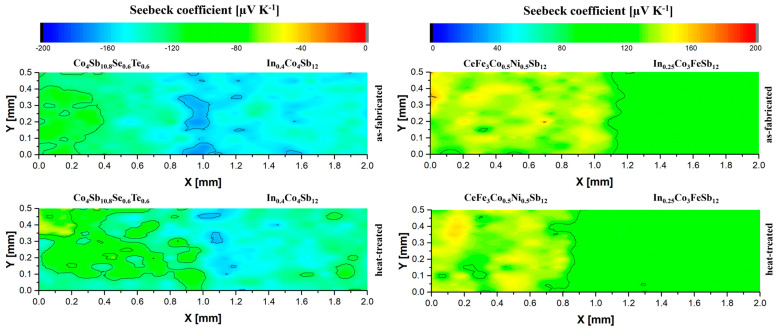
Seebeck coefficient maps of *n*- and *p*-type joints in as-fabricated and heat-treated states.

**Figure 11 materials-18-02923-f011:**
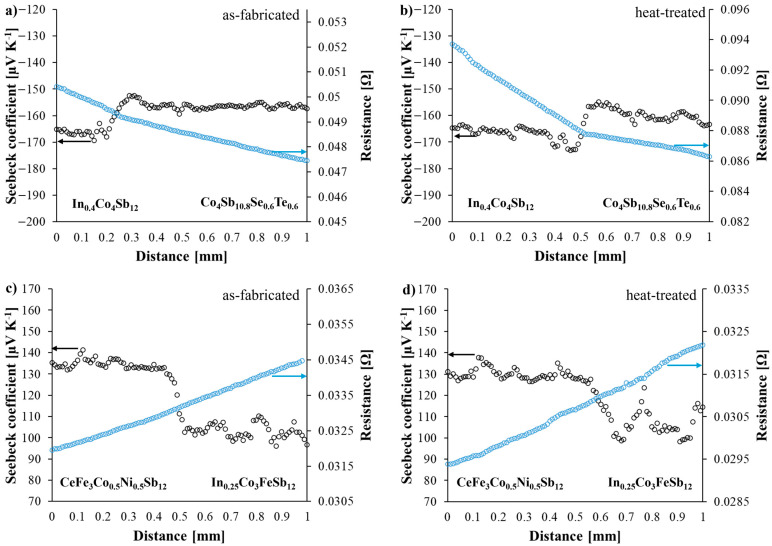
Seebeck coefficient and electrical resistance line scans of *n*- and *p*-type joints in as-fabricated and heat-treated states. (**a**) as-fabricated In_0.4_Co_4_Sb_12_/Co_4_Sb_10.8_Te_0.6_Se_0.6_ (**b**) heat-treated In_0.4_Co_4_Sb_12_/Co_4_Sb_10.8_Te_0.6_Se_0.6_ (**c**) as-fabricated CeFe_3_Co_0.5_Ni_0.5_Sb_12_/In_0.25_Co_3_FeSb_12_ (**d**) heat-treated CeFe_3_Co_0.5_Ni_0.5_Sb_12_/In_0.25_Co_3_FeSb_12_ joints.

**Table 1 materials-18-02923-t001:** Electrical contact resistance ρ_c_ of the joints.

Materials	Type	ρ_c_ [10^−6^ Ω cm^2^]	
		As-Fabricated	Heat-Treated
In_0.4_Co_4_Sb_12_/Co_4_Sb_10.8_Te_0.6_Se_0.6_	*n*-type	5.8 ± 1.1	4.4 ± 1.3
CeFe_3_Co_0.5_Ni_0.5_Sb_12_/In_0.25_Co_3_FeSb_12_	*p*-type	1.3 ± 0.3	1.0 ± 0.4

**Table 2 materials-18-02923-t002:** Measured CTE values of the thermoelectric materials at 310–723 K.

Material	Type	CTE [10^−6^ K^−1^]
In_0.4_Co_4_Sb_12_	*n*-type	10.6
Co_4_Sb_10.8_Te_0.6_Se_0.6_	*n*-type	11.7
CeFe_3_Co_0.5_Ni_0.5_Sb_12_	*p*-type	12
In_0.25_Co_3_FeSb_12_	*p*-type	10.5

**Table 3 materials-18-02923-t003:** Results of a calculation of the theoretical energy conversion efficiency η.

Material	ZT Max	η Max [%]	ZT Avg	η Avg [%]
Bulk legs				
In_0.4_Co_4_Sb_12_	1.18	15	0.93	12.9
Co_4_Sb_10.8_Te_0.6_Se_0.6_	1.1	14.4	0.66	10.1
CeFe_3_Co_0.5_Ni_0.5_Sb_12_	0.84	12.1	0.64	9.9
In_0.25_Co_3_FeSb_12_	0.89	12.6	0.63	9.7
Segmented legs				
In_0.4_Co_4_Sb_12_/Co_4_Sb_10.8_Te_0.6_Se_0.6_			0.96	13.2
CeFe_3_Co_0.5_Ni_0.5_Sb_12_/In_0.25_Co_3_FeSb_12_			0.66	10.1

## Data Availability

The raw data supporting the conclusions of this article will be made available by the authors on request.

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
