# Peer review of "Thermal and Interfacial Stability of PPS-Fabricated Segmented Skutterudite Legs for Thermoelectric Applications"

_materials, 2025, doi:10.3390/ma18132923_

Round 1
Reviewer 1 Report
Comments and Suggestions for Authors
In this paper, two joint materials, In0.4Co4Sb12 / Co4Sb10.8Te0.6Se0.6 (Type n) and CeFe3Co0.5Ni0.5Sb12 / In0.25Co3FeSb12 (Type p), were prepared by the pulsed plasma sintering method. The microstructure was studied by scanning electron microscopy and energy spectrum analysis. The electrical stability was measured by surface Zebeck coefficient mapping and contact resistance. The theoretical energy conversion efficiency of segmented N-type and P-type modules was calculated using the average ZT value. The results show that the prepared material combination has excellent thermal stability, chemical stability, and electrical stability. The research in this article has certain research significance and is worthy of being recommended for publication, but there are some problems that need to be revised.
1. The title of this article does not convey the research content of this article and needs to be revised and condensed to represent the research characteristics of this article.
2. In the abstract of this article, for the abbreviations used for the first time, the full names should be provided, such as ZT, XRD, SEM, and EDS, etc., and the full text should be checked.
3. In the introduction part, the author wrote too little content, and only listed the relevant literature without pointing out the existing problems and shortcomings. Therefore, a discussion of the existing literature is needed. In addition, 7 articles were cited, which is too few. Generally, it needs to be increased to 25. Please have the author add more relevant references to indicate that the author has not paid attention to the project's literature; otherwise, it will be difficult to be convincing.
4. In the "Materials and Methods" section, the author's introduction to the sample preparation is relatively simple. The preparation process of pulsed plasma sintering is not mentioned in this article. Please supplement. In addition, the author has briefly explained the microstructure, thermoelectric properties, and thermal stability tests of the materials. Please divide it into another paragraph for a detailed explanation.
5. The measurement of the coefficient of thermal expansion was not seen in the text. Please introduce the result of this measurement.
6. The author used the ZT value to calculate the theoretical energy conversion efficiency of N-type and P-type outriggers. What is the basis for this? Why are the data only 13.2% and 10.1%, respectively? Can it be further improved? Why?
7. The conclusion of this article should be described point by point and should not be summarized in separate paragraphs, which is not convenient for readers to review. Please modify it.
8. Most of the literature in the list is from 5 to 10 years ago, while there are relatively few from recent years. Considering the hot topics of this research, more attention should be paid to the literature from recent years, and thus, it needs to be revised.
Reviewer 2 Report
Comments and Suggestions for Authors
The manuscript is well written and logically structured. As a single-authored paper, it represents a commendable effort. However, the content appears unnecessarily prolonged at times, which may cause readers to lose focus and interest.
The following points require clarification:
-
The primary focus of the manuscript is the thermal stability study of segmented leg thermoelectric materials. However, the reference to the theoretical energy conversion value in the abstract (Page 1, Line 17) seems out of place. It may mislead readers into expecting a different focus. The abstract should be revised to emphasize only the author's original contributions or connect the fact to the following text.
-
The manuscript repeatedly refers to a specific duration of 168 hours for thermal stability testing (e.g., Page 5, Line 180). The significance of this duration is not explained. If this is a standard practice in the field, the author should provide a proper citation to support its use.
-
In Table 3 p.15, the values for segmented legs are identified as theoretical. Given that these values closely match those of the individual materials, the advantages of using segmented materials are not evident. This issue needs to be explicitly discussed.
- Since the figure of merit (ZT) depends on electrical conductivity, thermal conductivity, and the Seebeck coefficient, it inherently involves significant uncertainty. However, the manuscript does not mention any uncertainty values associated with the reported ZT. This omission should be addressed. This is because if the uncertainties are taken into account, it appears there may be no measurable improvement, as the values provided in Table 3, p.15.
